# Response to Salt Stress of the Halotolerant Filamentous Fungus *Penicillium chrysogenum* P13

**DOI:** 10.3390/molecules30061196

**Published:** 2025-03-07

**Authors:** Lyudmila Yovchevska, Jeny Miteva-Staleva, Vladislava Dishliyska, Galina Stoyancheva, Yana Gocheva, Radoslav Abrashev, Boryana Spasova, Maria Angelova, Ekaterina Krumova

**Affiliations:** 1Departament of Mycology, The Stephan Angeloff Institute of Microbiology, Bulgarian Academy of Sciences, Acad. G. Bonchev str. bl.26, 1113 Sofia, Bulgaria; lyovchevska@microbio.bas.bg (L.Y.); j_m@microbio.bas.bg (J.M.-S.); dishliyskav@microbio.bas.bg (V.D.); rabrashev@microbio.bas.bg (R.A.); bkasovska@abv.bg (B.S.); mariange@microbio.bas.bg (M.A.); 2Departament of General Microbiology, The Stephan Angeloff Institute of Microbiology, Bulgarian Academy of Sciences, Acad. G. Bonchev str. bl.26, 1113 Sofia, Bulgaria; galinadinkova@microbio.bas.bg (G.S.); yana2712@gmail.com (Y.G.)

**Keywords:** halophilic and halotolerant fungi, oxidative stress, biomarkers

## Abstract

In recent years, there has been increasing interest in the study of extremophilic microorganisms, which include halophiles and halotolerants. These microorganisms, able to survive and thrive optimally in a wide range of environmental extremes, are polyextremophiles. In this context, one of the main reasons for studying them is to understand their adaptative mechanisms to stress caused by extreme living conditions. In this paper, a fungal strain *Penicillium chrysogenum* P13, isolated from saline soils around Pomorie Lake, Bulgaria, was used. The effect of elevated concentrations of sodium chloride on the growth and morphology as well as on the physiology of the model strain was investigated. *P. chrysogenum* P13 demonstrated high tolerance to NaCl, showing remarkable growth in liquid and agar media. In order to establish the relationship between salt- and oxidative stress, changes in the cell biomarkers of oxidative stress, such as oxidatively damaged proteins, lipid peroxidation, and levels of reserve carbohydrates of the studied strain were evaluated. The involvement of antioxidant enzyme defense in the adaptive strategy of the halotolerant strain against elevated NaCl concentrations was investigated.

## 1. Introduction

The study of extremophilic microorganisms has gained increasing research interest. Halophilic microorganisms as part of extremophiles, are defined as surviving and multiplying at elevated NaCl levels, and for some, this is an obligatory condition for their development. These microorganisms, which can survive and thrive optimally under a variety of extreme environmental factors, are polyextremophiles [1,2]. Their tolerance parameters and salt requirements depend on specific environmental factors such as temperature, pH, and growth medium. Thus, one of the main reasons for their research is to elucidate the adaptive mechanisms to the stress resulting from harsh living conditions. They are classified as halotolerant (0.2–0.85 mol/L NaCl), moderate halophiles (0.85–3.4 mol/L NaCl), and extreme halophiles (3.4–5.1 mol/L NaCl) in dependence on their salt requirements to grow [3].

The common characteristics of halophilic microorganisms are low nutrient requirements and resistance to high salt concentrations as well as the ability to balance the osmotic pressure of the environment [4]. Halophilic microscopic fungi, as eukaryotes, attract research interest. In a saline environment, halophilic fungi are mainly subjected to two different types of stress—osmotic and ionic [5]. Hyperosmotic stress is characterized by a decrease in turgor pressure and dehydration of the cytoplasm, which leads to an increase in the concentration of solutes in the cytosol. Ionic stress is associated with the entry of ions into the cytoplasm due mainly to K+ ions. The increased concentration of Na+ ions inside the cells has a toxic effect on some cellular components, such as intracellular membranes and enzymes, and they are not part of the adaptation mechanisms [6,7].

A feature of salt adapted microorganisms is their ability to maintain a balance between the high osmotic pressure of the environment and the low water activity (wa) outside the cell, as opposed to that inside the cell (for prokaryotes) or intercellular (for eukaryotes) [3]. In order to maintain the sodium balance in the cytoplasm and counteract the osmotic pressure of the external environment caused by high salinity, their haloadaptive mechanisms include the intracellular storage of KCl above 37% (5 M) or the accumulation of compatible solutes [8]. Oxidative stress is another important challenge for microorganisms in saline habitats. As a result of stressful environmental conditions, the electron transport chains in the mitochondria of their cells are disrupted, leading to reverse electron flow and unwanted oxidation of oxygen by complex I and, consequently, the formation of reactive oxygen species (ROS) [9]. Some studies have shown that improved resistance to oxidative stress is linked to the acquisition of salt tolerance [10].

Salt adapted fungi possess the necessary mechanisms to balance cellular osmotic pressure and ion concentration, to stabilize cell membranes and to neutralize intracellular oxidative stress. They usually grow slowly as large amounts of energy are directed to the cellular mechanisms required for survival in adverse conditions [11]. Their survival at high salinity is the result of adaptive mechanisms at the cellular genetic, enzymatic and metabolic levels.

Published reports on halophiles in the scientific literature refer mainly to bacteria, while information on filamentous fungi is relatively scarce.

Saline habitats in Bulgaria are poorly studied. The Black Sea, salt lakes, and salt pans are unique saline ecosystems on the territory of Bulgaria and marine flora and fauna have been the subject of scientific interest for many years. The Bulgarian maritime space covers the western part of the Black Sea and the area of the sector is approximately 35,600 km^2^. The salinity of the Black Sea is relatively low—about 17.5–18 ‰ at the surface, and in the salt pans it is almost twice as high and varies depending on the season. The information available on halophilic and halotolerant fungi in the coastal zone and open waters of the Black Sea, as well as the adjacent salt lakes, is scarce.

One of the main reasons for studying halophilic fungi is their remarkable tolerance characteristics as well as ecological and morphological adaptability, which are crucial in efforts to understand the limits of life on Earth and in the Universe. Elucidating the mechanisms of adaptation to halotolerance of halophilic and halotolerant filamentous fungi would contribute to obtaining new knowledge necessary for the development of exobiology, which studies the origin, evolution, and spread of life in the Universe [12,13].

The majority of the available scientific data is focused on the adaptive mechanisms of halophiles and halotolerants living in conditions of hypersaline stress. There are few published data concerning the relationship between salt and oxidative stress in filamentous fungi. This article aims to investigate the response of a halotolerant fungus from the Bulgarian Black Sea region to increased salinity of the environment at the morphological and cellular level, as well as to obtain evidence for the relationship between salt and oxidative stress. The model strain was selected on the basis of its broad halotolerance. It is able to grow in the presence of elevated concentrations of sodium chloride (up to 20%) and in the absence of salt. Elucidation of the relationship between salt and oxidative stress would provide new information on the survival strategy of these polyextremophiles.

Considering the different survival mechanisms of halophilic fungi and their ability to synthesize metabolites with unique properties, any studies on them would enrich the available scientific information on the problem.

## 2. Results

Strain *Penicillium* sp. P13 was isolated from a soil sample taken from the shallows of Pomorie Lake. The Pomorie Lake is located in east Bulgaria with coordinates 42.583° N. 27.618° E. It is used for salt production (about 30,000 t per year) and medicinal mud. The lake salinity is 60–80‰. We chose a lake from the Black Sea region because it is poorly studied for the presence of halophilic and halotolerant mycobiota. The *Penicillium* sp. P13 strain was chosen for the present study, due to its wide tolerance to sodium chloride. The strain was identified based on molecular genetic studies as *Penicillium chrysogenum* P13. Its sequence has been deposited in the gene bank of NCBI under accession number PQ726899.

### 2.1. Effect of Different Salt Concentrations on Growth and Glucose Consumption of the Halotolerant Fungal Strain

The effect of different salt concentrations on the *P. chrysogenum* P13 growth is shown in Figure 1. Presented data demonstrated the model strain capability to growth on agar medium within a NaCl range of 0–20%, with an optimum at of 5% NaCl. Although the strain was isolated on agar medium with 5% salt, it showed growth on medium supplemented with 5 to 20% sodium chloride and also grew well in the absence of salt. Based on these experiments, we classified the strain as moderate halotolerant.

A growth of the model strain is observed at all tested salt concentrations (Figure 1). The surface of the strain colony is velvety and green in color. The colony’s characteristic white ring is noticeable in all variants. Although the colony’s size decreases as salinity rises, sporulation is seen even at the maximum NaCl concentration (Figure 1).

The growth of the strain studied is observed in submerged cultivation conditions. In all the variants tested, we observed a smooth growth of the culture (Figure 2). The presence of elevated concentrations of sodium chloride in the medium causes a slowdown in the growth of the model strain in a dose-dependent manner. In the first three variants (0–5%), the development of the strain showed an identical trend, which confirms its belonging to the halotolerant fungi. The next concentrations used (7.5–10%) led to a reduction in the amount of biomass produced, but the growth curve is again typical for the filamentous fungi, albeit at a slower rate. Although biomass gradually increased over time, the levels were noticeably lower than those of the variant with lower NaCl concentrations, suggesting that the culture is under stress.

Glucose uptake is also affected by salt concentrations in the culture medium (Figure 3). The maximum amount of glucose is consumed by cultures incubated in the presence of 0–5% salt. The rapid and abundant growth of the studied halotolerant strain coincides with the accelerated consumption of glucose from the medium. Increasing the concentration of NaCl causes inhibition of the process. In the variants up to 5% salt, the delay in the consumption of the carbon source is insignificant. Figure 3 clearly shows that at 96 h from the start of cultivation, the levels of glucose consumption are identical to that of the variant cultivated in the absence of salt. The model strain cultured under high salt conditions (7.5 and 10% NaCl) showed delayed glucose consumption compared to the variants cultured in the presence of lower salt concentrations. This is probably one of the mechanisms of adaptation to the increased salinity of the environment.

### 2.2. Determination of the Oxidatively Damaged Proteins Level by the Amount of Carbonyl Groups

The changes in the amount of oxidatively damaged proteins in the cells of the model strain, cultured in the absence and presence of different concentrations of sodium chloride were investigated (Figure 4).

The carbonyl groups formed as the result of oxidative damage of cells were used as a marker of oxidatively damaged proteins. The results of our experiments showed a gradual increase in the amounts of oxidatively damaged protein molecules in all variants examined. The lowest contents of carbonyl groups were recorded in the variants with 2.5 and 5% sodium chloride, and the highest—in the variants with 7.5 and 10%. The strain cultivation in the absence of salt resulted in a strong and rapid increase in carbonyl group levels in the period from 48 to 72 h, and in the remaining periods the level stabilized.

### 2.3. Study of Changes in the Level of Reserve Carbohydrates–Glycogen and Trehalose

The bioindicators of oxidative stress include the level of reserve carbohydrates glycogen and trehalose. To investigate the involvement of these indicators in adaptation to salt stress, we subjected the studied fungal strain to prolonged exposure to different concentrations of sodium chloride.

The synthesis of glycogen and trehalose during cultivation of the model strain in the presence of different salt concentrations in the culture medium is shown in Figure 5. An increase in glycogen content is already observed in the first 24 h after the start of cultivation, and there are no significant differences in its levels in all variants tested (Figure 5A). In the following periods studied, a gradual increase in glycogen levels is observed, and again it is most significant in the variants in high salt concentrations. For the second reserve carbohydrate, an increase in its level is observed at 72 h, and again it is most significant in the variants with 7.5 and 10% NaCl. It should be noted that at 72nd hour the amount of trehalose in the *P. chrysogenum* P13 cells, cultured in the presence of high concentrations of sodium chloride is about 20 times higher compared to that at 48th hour (Figure 5B).

The accumulation of both reserve carbohydrates indicated a stressed state of the culture under these conditions. The results obtained confirm the essential role of trehalose and glycogen for the development of fungi isolated from extreme habitats. It is still unclear how reserve carbohydrates protect the cell from the negative consequences of stress. Trehalose is thought to reduce oxidative damage to lipids and proteins that would otherwise be degraded by ROS [14]. It slows down the rate of protein aggregate formation and its presence in the lipid bilayer is mandatory.

### 2.4. Study of Lipid Peroxidation as a Marker for the Degree of Membrane Integrity

Malondialdehyde (MDA) is a compound formed by the peroxidation of polyunsaturated fatty acids. Usually, the amount of MDA is used as a marker of lipid peroxidation in cells. In the next experiments, a significantly higher amount of MDA was recorded as the NaCl concentration in the culture medium increased (Figure 6). This increase depends on the degree of the stress factor and increases proportionally to the duration of cultivation under salt stress conditions. Interestingly, the accelerated accumulation of damaged lipids is observed after 72 h from the start of cultivation, with the lowest levels being recorded when the cells of the fungal strain are exposed to concentrations of 2.5 and 5% NaCl, probably due to its adaptation to the conditions of the environment from which it was isolated.

The increased levels of lipid peroxidation observed are due not only to direct lipid damage but also to the increased amount of hydrogen peroxide as a result of the stress. This, in turn, accelerates the Fenton and Haber-Weiss reactions, which leads to an increase in the level of hydroxyl radicals responsible for lipid peroxidation [15,16]. However, there is little information on the relationship between saline habitats and oxidative stress in filamentous fungi.

The cellular response to oxidative stress also includes induction of the antioxidant enzymes superoxide dismutase (SOD) and catalase (CAT). We monitored changes in the activity of both antioxidant enzymes in *P. chrysogenum* P13 cells in the absence and presence of sodium chloride, as well as the activities of the extracellular forms of the enzymes.

An increase in the activity of extracellular SOD was observed as early as 24 h after the contact of the fungal cells with NaCl (Figure 7A). No significant differences in the reported activity of each variant have been observed over time. Variations in activity were observed between the variants studied, with the highest activity observed in the variants with the highest sodium chloride concentrations used.

The activity of intracellular SOD is significantly higher than that of the extracellular enzyme (Figure 7A,B). Again, stimulation of SOD activity is observed as early as 24 h and it gradually increases until the end of the experiment. The observed trend here is the same as for the other stress biomarkers. The lowest levels of enzyme activity are observed in the variants with 2.5 and 5% salt and the highest—at 7.5 and 10% salt. In the variant using the maximum salt concentration, the enzyme activity increased almost 3 times from 24th to 96th hour of cultivation. The increase in SOD activity is probably due to an increase in the amount of superoxide anion radical (O_2_) under salt stress conditions.

The changes in the activity of the second antioxidant enzyme catalase showed the similar trend. Figure 8 shows increased extracellular enzyme activity as cells response to the increased concentration of salts in the medium. Already at 24th hour, increased enzyme activity is observed in a dose-dependent manner and this trend persists up to 96th hour. At the high concentrations used, the enzyme activity in the 96th hour is almost ten times higher than in the variant without salt.

Stimulation of the intracellular enzyme activity is also reported at 24th hour. As can be seen from Figure 9, there is no significant difference in the reported enzyme activities of all the variants studied at the 24th hour. In the variant with 0% salt, the change in catalase activity from 24th to 96th hour is insignificant.

The following variants result in a slight increase in activity over time. The strongest stimulation of intracellular catalase activity is observed in the 72nd and 96th hours in the variants with higher salt concentrations—7.5 and 10% NaCl. This increase is three times higher for the 7.5% variant compared to the activity recorded after 24 h and about four times higher for the 10% variant.

From the experiments carried out related to the activity of antioxidant enzymes, it can be seen stimulation of the enzymatic antioxidant protection of the cells of the halotolerant strain, which occurs under conditions of high salt content—7.5 and 10%, i.e., cultivated under conditions of salt stress. This increase is due to the formation of superoxide radicals as a result of salt stress and subsequently increased H_2_O_2_ levels.

## 3. Discussion

In the present study, a newly isolated strain from a poorly studied saline ecological niche was used. The strain was isolated from a saline habitat with a salinity of 6–8%. Based on its growth and morphology in the presence of different concentrations of NaCl, we can classify it as moderate halotolerant. As can be seen from Figure 1 and Figure 2, its development is very good both in the presence of 2.5 and 5% salt and in its absence. Growth and development of the strain are observed even in the presence of 20% sodium chloride, which indicates its adaptation to life and development at high salt concentrations in the environment.

Based on the ITS rDNA region, the strain was identified as *Penicillium chrysogenum* P13. *Penicillium* is one of the dominant fungal genera inhabiting saline habitats. Many different *Penicillium* species have been reported from saline habitats (hypersaline areas, saline soils, saline seawater, lakes): isolates of *Aspergillus* and *Penicillium* are dominant, while *Cladosporium* is less abundant in halophilic mycobiota from the Dead Sea. *Penicillium* is also one of the dominant genera among halophilic fungal isolates from man-made solar salt pans in Pattani Province, Thailand [17,18].

Some *Aspergillus* species reported are facultative halophiles, while *Cladosporium* and *Penicillium* species are exclusively facultative halophiles [19]. Identification of isolated halophilic fungi from different geographical regions has proven the presence of fungi from the genera *Aspergillus*, *Penicillium*, *Alternaria*, and *Cladosporium* [20]. Many of them show xerotolerance and grow well under low wa conditions. Species from the genera *Aspergillus* and *Penicillium* are prevalent in the mangrove forests and salt marshes of Goa, India. They also have high resistance to salts of heavy metals Pb^2+^, Cu^2+,^ and Cd^2+^ [21].

Under high salt stress, cells undergo physiological responses such as osmotic pressure imbalance, metabolic disorders, oxidative stress, and ionic toxicity, and fungal cells are no exception. Salinity-induced osmotic stress and ionic stress trigger the overproduction of ROS and ultimately lead to oxidative damage to cell organelles and membrane components, and at severe levels cause cell and plant death [22]. When exposed to high salt conditions, stress conditions disrupt the electron transport chain in mitochondria, leading to reverse electron flow and unwanted oxygen oxidation from complex I, respectively, to increased generation of reactive oxygen species (ROS) [9]. Some studies in plants suggest that the acquisition of salt tolerance may be the result of improved resistance to oxidative stress [23]. To confirm the relationship between oxidative and salt stress, we investigated some essential stress biomarkers.

It is assumed that microorganisms adapted to adverse conditions must synthesize a wide range of compounds necessary for their survival such as reserve carbohydrates, exopolysaccharides, polyols, melanins, etc. The synthesis of all these compounds affects the growth rate of fungi. Our results showed that under salt stress conditions, fungal growth is slowed down compared to variants cultivated in the presence of the lower concentrations of NaCl, as well as in the absence of salt (Figure 2). Cells glucose uptake illustrates a clear dependence on the degree of salinity (Figure 3). We found a significant decrease in glucose consumption in *P. chrysogenum* P13 under conditions of increased salinity of 7.5 and 10% salt. These data can be considered as a confirmation of the above-mentioned assumption of differences in biomass production under salt stress. Increasing concentration of NaCl leads to a significant decrease in glucose uptake. Thus, salinity-induced stress may cause a dynamic rearrangement of the metabolic flux towards the pentose phosphate pathway, leading to the generation of the reduced electron carrier NADPH.

As a marker for oxidatively damaged protein molecules, we use the carbonyl groups formed as a result of oxidative damage to proteins. Our results showed that protein molecules in the studied halotolerant strain exhibit resistance to salt concentrations up to 5%. Further, an increase in the salinity of the medium to 7.5 and 10% led to manifestations of oxidative stress and an increase in the level of oxidatively damaged protein molecules (Figure 4).

The production of reserve carbohydrates by stressed cells appears to be a critical adaptation that protects microorganisms from a wide variety of potentially lethal conditions [24,25]. Trehalose is a reducing disaccharide, acting as a protectant against various environmental stresses in numerous organisms. Changes in trehalose levels under salt stress conditions have been studied in the plant *Anoectochilus roxburghii* [26]. Glycometabolism products, including trehalose, show marked changes under NaCl stress and their genes are expressed in response to NaCl stimulation and play a key role in the metabolism of polysaccharides and glycosides in *Anoectochilus* [26]. The role of trehalose in increasing the resistance of organisms to the damaging effects of salt stress has also been established in the tomato plant [27]. An increase in glycogen levels has been found and published in filamentous fungi subjected to various types of stress, such as cold stress, stress caused by heavy metals, etc. [28,29]. The importance of the reserve carbohydrates glycogen and trehalose in increasing resistance to salt stress has been documented primarily in studies on plant objects. Our results related to an increase in the reserve carbohydrates glycogen and trehalose prove and confirm their role in increasing the resistance and survival of filamentous fungi placed in conditions of increased salinity (Figure 5).

The biomarkers of oxidative stress MDA and protein oxidation caused by increased salt content were studied on the *Aspergillus sydowii* EXF-12860. Saturated concentrations of NaCl induce oxidative stress in *A. sydowii*. The authors of the study found that the increase in the levels of the studied biomarkers indicates the presence of oxidative stress, which induces a cellular response [30]. This cellular response includes enzymatic and non-enzymatic components. The increased content of MDA in the cells of our model strain, cultured in the presence of high concentrations of sodium chloride, confirms the presence of oxidative stress (Figure 6). Exposure of microorganisms to stress leads to increased production of reactive oxygen species in the cell. To limit the resulting damage, cells have evolved various antioxidant defenses. Whole-genome sequencing of some of the most halotolerant and halophilic fungal species allows the investigation of relationships between oxidative and salt stress tolerance at the genomic level. Gostiňcar and Gunde-Cimerman [31] indicate that the maximum tolerated salinity correlated with the expression of genes encoding three major enzymes of the cellular oxidative stress response: superoxide dismutases, catalases, and peroxiredoxins. This observation supports the hypothetical link between the antioxidant capacity of cells and their halotolerance [31]. Jiménez-Gómez, et al. [30] described for the first time non-enzymatic and enzymatic antioxidant defenses in filamentous fungi exposed to hypersaline conditions, thus increasing the knowledge of interactions between halophilic fungi and NaCl [30].

Catalase, superoxide dismutase, and glutathione reductase together act as a primary antioxidant enzyme defense. *A. sydowii* triggers a broad antioxidant response against cellular oxidative damage. The transcriptome data from this study showed that genes involved in different cellular antioxidant defenses are expressed under salt stress conditions. Other fungi that induce antioxidant metabolites—enzymatic and non-enzymatic—have also been reported to respond to osmotic stress [31]. Stimulation of enzymatic antioxidant defenses has also been found in *Aspergillus cristatus* cells subjected to salt stress [32]. Biochemical data indicate that catalase, superoxide dismutase, and glutathione reductase together act as a major antioxidant defense system to protect *A. sydowii* cells against peroxidative molecules under extreme salinity conditions [31].

*Aspergillus loretoensis* also expresses genes such as SOD2 (oxidative stress and oxygen toxicity), ASG (salinity tolerance) and transmembrane transport as a response against the presence of NaCl in the medium. At the same time, genes up-regulated at elevated NaCl concentrations (15%) are associated with osmolyte transport, carbohydrate transport, and metabolism, all of which suggest their adaptive properties [33].

The antioxidant defense system also protects plants from salt-induced oxidative damage by detoxifying ROS and also by maintaining the balance of ROS generation under salt stress [22]. Experiments with ryegrass under salt stress have also reported an increase in antioxidant enzyme activity. Neither the salt treatment nor the sulfur nanoparticles (SNP) affected photosynthetic rate (Fv/fm), but both treatments induced glutathione content. SNP treatment restored the elevated activities of CAT, SOD, peroxidase, ascorbate peroxidase and polyphenol oxidase caused by salt stress [34]. The obtained results from studies with pearl millet seedlings show that transcriptional regulation of genes can increase the activity of cells antioxidant enzymes. Based on this finding it is possible to build a strategy to protect pearl millet seedlings from salt stress by reducing oxidative stress [35].

The increased activity of cells antioxidant enzymes of *Penicillium chrysogenum* P13 in accordance with the increased salinity of the environment (Figure 7, Figure 8 and Figure 9) confirms their main role in the processes of adaptation and survival of filamentous fungi in conditions of high salinity.

All of the above provides further evidence for the relationship between oxidative and salt stress in a model phylamentous fungus. Increased levels of cellular biomarkers of oxidative stress clearly correlate with increased environmental salinity. The new information contributes to the elucidation of cellular mechanisms of adaptation to salt stress.

## 4. Materials and Methods

### 4.1. Experimental Material

In this work, filamentous fungi from the Mycological Collection of the Institute of Microbiology, Bulgarian Academy of Sciences were used. For their isolation, soil samples (3 cm deep below the top layer) were used, which were collected sterilely in the summer of 2022 from Pomorie Lake. Each sample contains 10 g of soil, randomly collected in sterile tubes and stored at −20 °C.

Pomorie Lake is a coastal lagoon located north of the town of Pomorie. Its area is about 7–8.5 km^2^, its length is 5–6 km, and its width varies from 350 m in the north to 1.6 km in the middle part. Its depth does not exceed 1.4 m, and its salinity is 60–80‰. It is used for salt production (about 30,000 tons per year) and medicinal mud.

### 4.2. Molecular Genetic Identification of the Species Level of the Isolates

Molecular genetic identification of the selected strain was carried out based on sequencing of the ITS rDNA region, as described Stoyancheva et al. [36].

### 4.3. Nutrient Media and Culture Conditions

#### 4.3.1. Solid Nutrient Media

To study the changes in the morphology of the studied strain, cultivation was carried out superficially in Petri dishes on potato dextrose agar in the absence or presence of different concentrations of sodium chloride (0; 2.5; 5; 7.5; 10%). Cultivation was carried out in a thermostat at 25 °C for 10 days.

#### 4.3.2. Liquid Culture Media

To determine the cellular changes resulting from the effects of salt stress, depth cultivation was carried out in 500 mL Erlenmeyer flasks. Five different variants were used using potato dextrose broth as a medium with the addition of different concentrations of sodium chloride (0; 5; 7.5; 10%). Cultivation was carried out on a shaker at a temperature of 25 °C, with samples taken at 24, 48, 72, and 96 h from the start of cultivation.

All experiments were performed in triplicate.

### 4.4. Analytical Methods

#### 4.4.1. Morphological and Physiological Measurements

The growth of the fungal culture was determined by measuring the dry weight of the formed biomass. Mycelium from a certain phase of culture development was filtered through a Whatman (Clifton, NY, USA) No. 4 filter, washed with distilled water, and dried at 105 °C to constant weight.

#### 4.4.2. Cell-Free Extract Preparation

The cell-free extract was prepared as described earlier [37]. Biomarkers of oxidative stress are determined in the resulting cell-free extract.

### 4.5. Enzyme Activities Determination

The specific activity of SOD (EC 1.15.1.1.) is determined by the method of Beauchamp and Fridovich [38]. One unit of SOD activity (expressed as U/mg protein) is the amount of enzyme protein required to inhibit the rate of NBT reduction (A560) by 50% at pH 7.8 and temperature 30 °C.

The specific activity of catalase (EC 1.11.1.6.) is determined by the method of Beers and Sizer [39]. One unit of activity is the amount of enzyme protein required to degrade 1 mM H_2_O_2_ for 1 min at 25 °C and pH 7.0.

### 4.6. Other Determinations

#### 4.6.1. Soluble Reducing Sugars Determination

Soluble reducing sugars were determined by the Somogy–Nelson method (Somogy, 1952), with glucose as a standard [40].

#### 4.6.2. Total Protein Determination

The amount of protein was measured by the method of Lowry et al. (1951), using bovine serum albumin as a standard [41].

#### 4.6.3. Determination of the Content of Oxidatively Modified Proteins

Determination of the content of carbonyl groups in oxidatively modified proteins is performed according to the method of Adachi & Ishii [42].

#### 4.6.4. Determination of the Content of Reserve Carbohydrates

The content of glycogen and trehalose was determined by a procedure described by Becker (1978) and Vandercammen (1989) and modified by Parrou et al. [43,44,45]. The released glucose was determined by the Somogy–Nelson method (Somogy, 1952) [40].

#### 4.6.5. Determination of the Lipid Peroxidation Level

Determination of lipid peroxidation was performed using a kit from Sigma-Aldrich (Burlington, MA, USA).

All the experiments were performed in triplicate.

### 4.7. Statistical Evaluation of the Results

The results obtained in this investigation were evaluated from at least three repeated experiments using three parallel runs and reported values to represent the mean. The error bars indicate the standard deviation (SD) of the mean of triplicate experiments. The data were analyzed using one-way analysis of variance (ANOVA), followed by Tukey’s test. For the statistical processing of the data, the version of the ANOVA software built into the Origin program (OriginPro 2019b, 64-bit) was used.

## 5. Conclusions

The present study presented the tolerance of a newly isolated strain from a poorly studied ecological niche to different concentrations of NaCl. The strain was identified as *Penicillium chrysogenum* P13. The strain’s adaptations to development and survival in the presence of different salt concentrations were studied. An increase in the levels of the studied biomarkers of oxidative stress was found in the cultivation of the strain in the presence of high concentrations of NaCl (7.5 and 10%), which confirms the state of oxidative stress in which the fungal cells are located. The relationship between salt and oxidative stress in a filamentous fungus was confirmed.

## Figures and Tables

**Figure 1 molecules-30-01196-f001:**
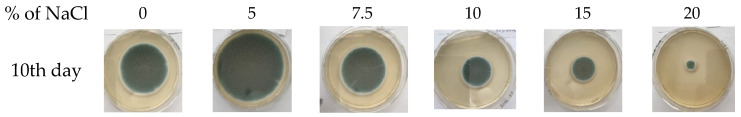
Colony growth of *P. chrysogenum* P13 on PDA medium in the presence of enhanced concentrations of NaCl.

**Figure 2 molecules-30-01196-f002:**
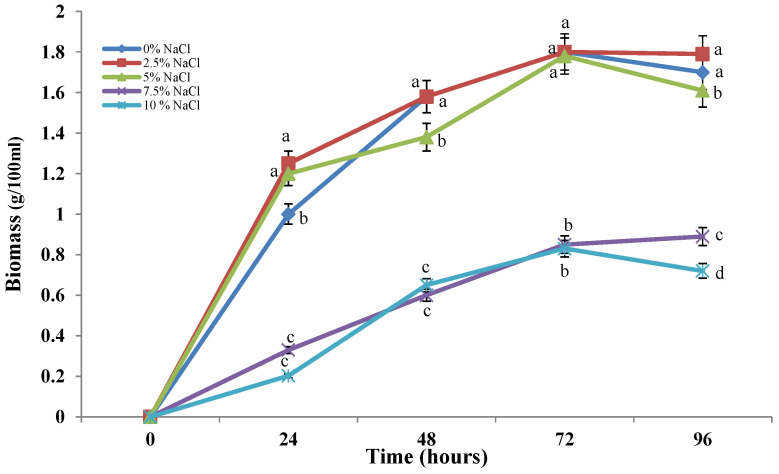
Effect of NaCl concentrations on biomass production by *P. chrisogenum* P13. Values are the means of three replicate experiments with three replicates in each experiment; bars represent the standard deviation. Different lower letters indicate significant differences (Tukey’s test *p* < 0.05) between salt stress exposure at the same cultivation period.

**Figure 3 molecules-30-01196-f003:**
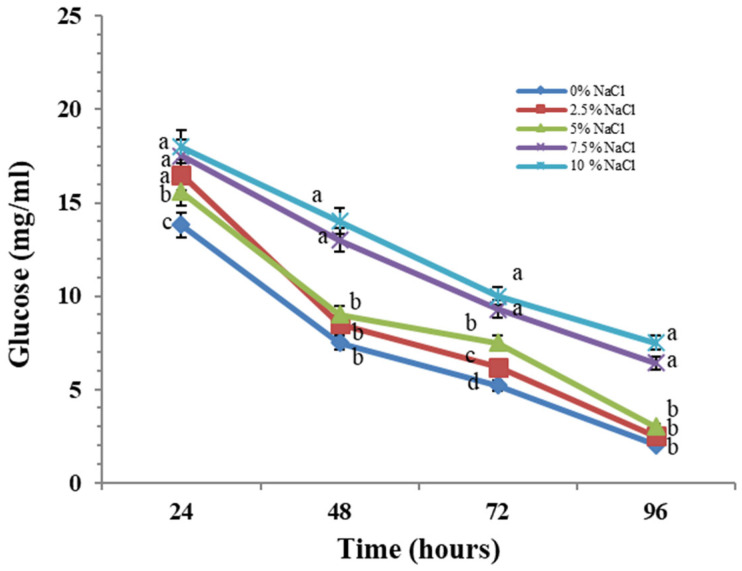
Glucose consumption of *P. chrisogenum* P13 cells in the presence of enhanced concentrations of NaCl. Values are means of three repeated experiments with three replicates in each trial; bars represent the standard deviation. Different lower letters indicate significant differences (Tukey’s test *p* < 0.05) in glucose consumption between salt stress treatments at the same cultivation period.

**Figure 4 molecules-30-01196-f004:**
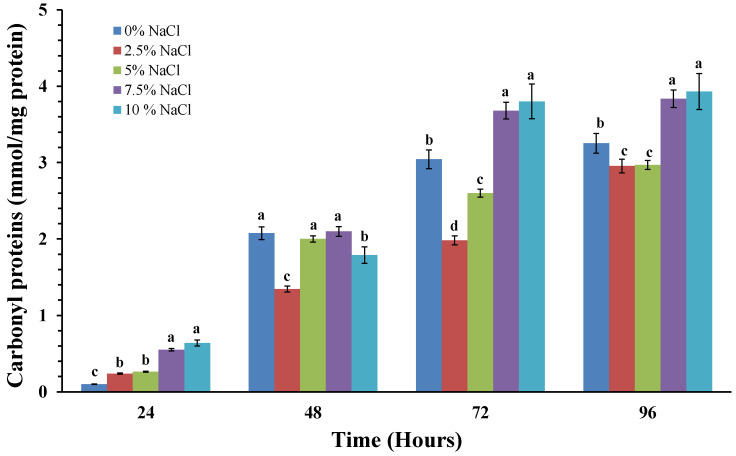
Effect of NaCl on protein carbonylation in *P. chrisogenum* P13 cells. Increased concentrations of NaCl had a statistically significant effect on oxidatively damaged proteins (Tukey’s test, *p* > 0.05). Different lower-case letters indicate significant differences between salt stress treatments at the same cultivation period.

**Figure 5 molecules-30-01196-f005:**
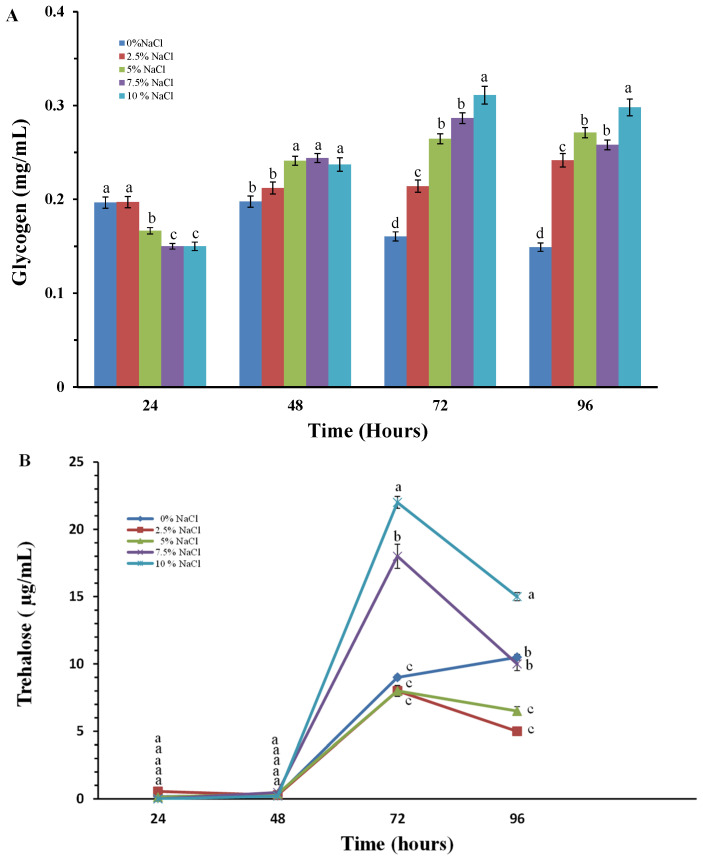
Glycogen (**A**) and trehalose (**B**) content in the *P. chrisogenum* P13 cells after exposure to elevated NaCl concentrations. Values are means of three repeated experiments with three replicates in each trial; bars represent the standard deviation. Different lower-case letters indicate significant differences (Tukey’s test *p* < 0.05) between salt stress treatments at the same cultivation period.

**Figure 6 molecules-30-01196-f006:**
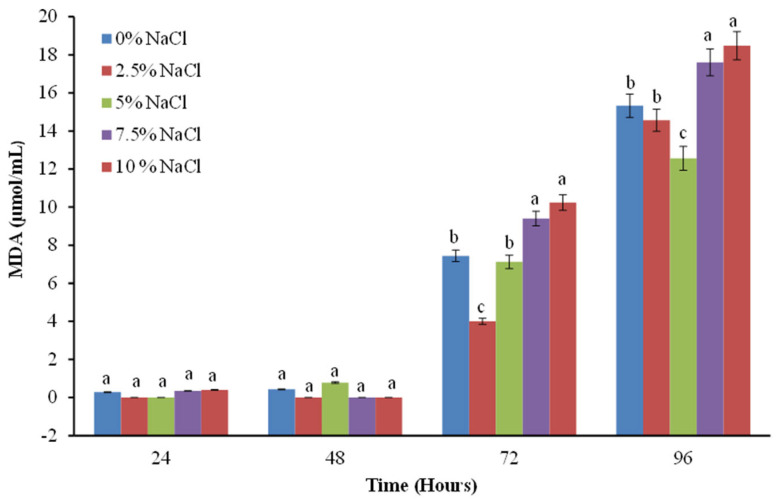
Levels of MDA in *P. chrisogenum* P13 cells as a result of NaCl concentration exposure. Values are the means of three repeated experiments with three replicates in each trial; bars represent the standard deviation. Different lower-case letters indicate significant differences (Tukey’s test *p* < 0.05) between salt stress treatments at the same cultivation period.

**Figure 7 molecules-30-01196-f007:**
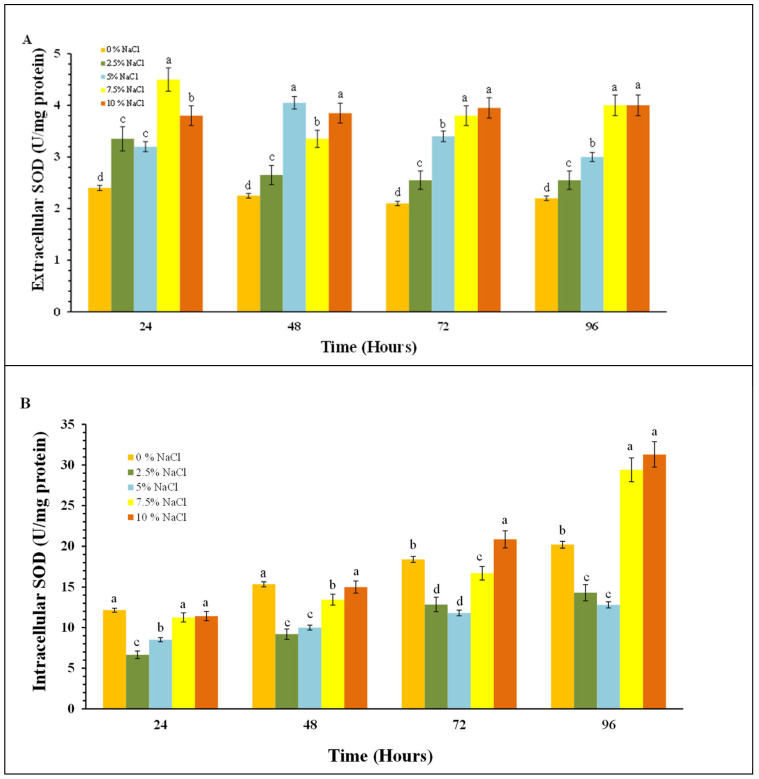
SOD activities in *P. chrisogenum* P13 cells as a function of NaCl concentration. (**A**) Extracellular SOD activity, and (**B**) intracellular SOD activity. Values are means of three repeated experiments with three replicates in each trial; bars represent the standard deviation. Different lower-case letters indicate significant differences (Tukey’s test *p* < 0.05) between salt stress treatments at the same cultivation period.

**Figure 8 molecules-30-01196-f008:**
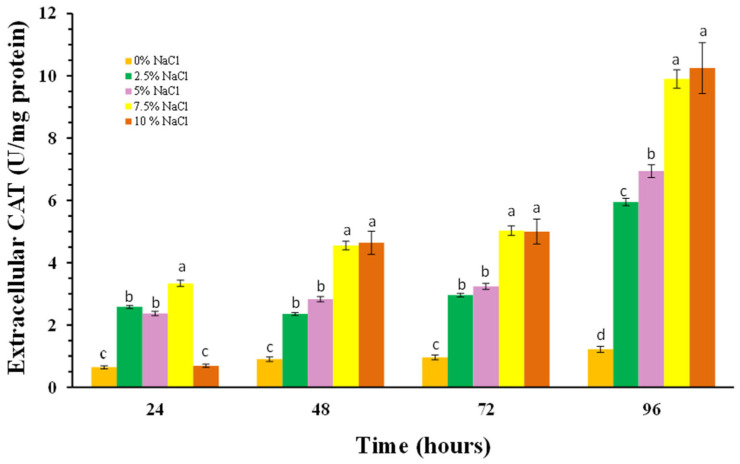
Extracellular CAT activities in *P. chrisogenum* P13 cells depending on concentrations of NaCl. Values are means of three repeated experiments with three replicates in each trial; bars represent the standard deviation. Different lower-case letters indicate significant differences (Tukey’s test *p* < 0.05) between salt stress treatments at the same cultivation period.

**Figure 9 molecules-30-01196-f009:**
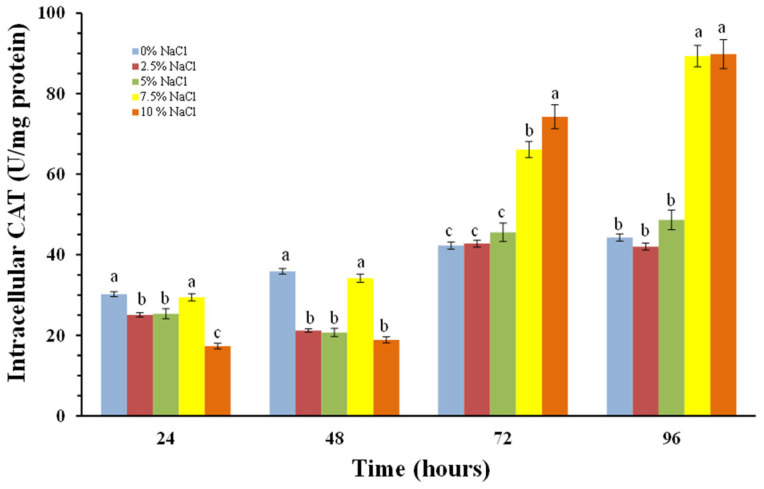
Intracellular CAT activity depends on NaCl concentrations in *P. chrisogenum* P13 cells. Values are means of three repeated experiments with three replicates in each trial; bars represent the standard deviation. Different lower-case letters (a–c) indicate significant differences (Tukey’s test *p* < 0.05) between salt stress treatments at the same cultivation period.

## Data Availability

Data are contained within the article.

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
