# Peer review of "Response to Salt Stress of the Halotolerant Filamentous Fungus Penicillium chrysogenum P13"

_molecules, 2025, doi:10.3390/molecules30061196_

Round 1
Reviewer 1 Report
Comments and Suggestions for Authors
This work is concerned the cellular response to salt stress of the halotolerant filamentous fungus Penicillium chrysogenum P13, isolated from a poorly studied ecological niche of Pomorie Lake, Bulgaria. The most attention was paid to response against oxidative stress. This work expands the knowledge of halophilic fungi adaptations and survival mechanisms in saline environments.
Before this manuscript is published, some important corrections need to be made. This work lacks statistical analysis of the results, although the authors in the methods section declared to perform it.
Are the observed differences in antioxidant enzyme activity, TBARS level, and other parameters studied statistically significant?
How many repetitions of experiments were performed?
What is meant by: “individual repetitions of experiments”? – line 443
Please describe correctly the species name of the tested fungus throughout the manuscript, i.e., Penicillium chrysogenum (not chrisogenum).
Discussion: needs to be restructured and shortened. Be concise: What's the main findings you would like to discuss, and what do the findings imply for further study? Please remove the description of the results obtained for various halophilic fungi without any relation to the results obtained in this work, e.g. the authors wrote: “It was found that maximum tolerated salinity correlated with the expression of genes encoding three major enzymes of the cellular response to oxidative stress: superoxide dismutases, catalase and peroxiredoxin.” So why wasn't peroxiredoxin activity examined in P. chrysogenum?
Please order the abbreviations used for the Latin names of fungal species, e.g., A. sydowii (Line 333) after the generic name Aneoectochlus (Line 323) suggests that A. sydowii is Aneoectochlus sydowii, whereas it is Aspergillus sydowii;
Line 309 – Hortaea werneckii (not H. werneckii)
Some names of genus and species are not in italics, e.g., lines 258, 307
Line 410 – It should be fungal culture; mushrooms are macroscopic fungi.
Line 91 – It should be mycobiota, not mycoflora
Line 266 – wa conditions, not aw
Improve figure descriptions:
Figure 2, express biomass in g/L;
Figure 6 , express MDA in µmol/ml - values from 2 to 20, instead of 2000 to 20000;
Figure 7 - 9, express SOD and CAT in U/mg protein.
Author Response
Dear Reviewer,
We appreciate all your valuable comments on our manuscript. We have revised the manuscript in accordance with the comments and suggestions. We believe that the manuscript has been further improved. You can find description of our corrections bellow.
General comments: This work is concerned the cellular response to salt stress of the halotolerant filamentous fungus Penicillium chrysogenum P13, isolated from a poorly studied ecological niche of Pomorie Lake, Bulgaria. The most attention was paid to response against oxidative stress. This work expands the knowledge of halophilic fungi adaptations and survival mechanisms in saline environments.
Comment 1. Before this manuscript is published, some important corrections need to be made. This work lacks statistical analysis of the results, although the authors in the methods section declared to perform it.
Are the observed differences in antioxidant enzyme activity, TBARS level, and other parameters studied statistically significant?
Response 1. Thank you for your advice! We added statistical analysis of the results
Comment 2. How many repetitions of experiments were performed?
Response 1. All the experiments were performed in triplicate. We add this in Materials and methods section.
Comment 3. What is meant by: “individual repetitions of experiments”? – line 443
Response 1. Sorry for the incorrected sentence. We mean separate repetitions of each experiment.
Comment 4. Please describe correctly the species name of the tested fungus throughout the manuscript, i.e., Penicillium chrysogenum (not chrisogenum).
Response 4. Sorry for this mistake! We corrected it!
Comments 5. Discussion: needs to be restructured and shortened. Be concise: What's the main findings you would like to discuss, and what do the findings imply for further study? Please remove the description of the results obtained for various halophilic fungi without any relation to the results obtained in this work, e.g. the authors wrote:
Response 5. We restructured and shortened the discussion following your advises.
Comment 6. “It was found that maximum tolerated salinity correlated with the expression of genes encoding three major enzymes of the cellular response to oxidative stress: superoxide dismutases, catalase and peroxiredoxin.” So why wasn't peroxiredoxin activity examined in P. chrysogenum?
Response 6. “Gostinˇcar, C.and Gunde-Cimerman, N indicate that the maximum tolerated salinity correlated with the expression of genes encoding three major enzymes of the cellular oxidative stress response: superoxide dismutases, catalases, and peroxiredoxins.”
In our experiments we choose to assay activity of SOD and CAT as enzymes of primary line of enzyme antioxidant defense.
Comment 7. Please order the abbreviations used for the Latin names of fungal species, e.g., A. sydowii (Line 333) after the generic name Aneoectochlus (Line 323) suggests that A. sydowii is Aneoectochlus sydowii, whereas it is Aspergillus sydowii;
Response 7. We corrected the mentioned abbreviations used for the Latin names of fungal species
Comment 8. Line 309 – Hortaea werneckii (not H. werneckii)
Response 8. Thank you for your advice! We removed this sentence following your advice for discussion focusing.
Comments 9. Some names of genus and species are not in italics, e.g., lines 258, 307
Response 9. Sorry for this miss! We corrected it.
Comment 10. Line 410 – It should be fungal culture; mushrooms are macroscopic fungi.
Response 10. Sorry for this mistake! We corrected the sentence!
Comment 11. Line 91 – It should be mycobiota, not mycoflora
Response 11. We corrected the sentence!
Comment 12. Line 266 – wa conditions, not aw
Response 12. We corrected it!
Comment 13. Improve figure descriptions:
Figure 2, express biomass in g/L;
Response 13. Thank you for your advice! We prefer to express the biomass in g/100 ml. We think the trend will be the same.
Comment 14. Figure 6 , express MDA in µmol/ml - values from 2 to 20, instead of 2000 to 20000;
Response 14. Thank you for your advice! We corrected the axis title and scale.
Comment 15. Figure 7 - 9, express SOD and CAT in U/mg protein.
Response 15. Thank you for your advice! We corrected the axis titles.
Reviewer 2 Report
Comments and Suggestions for Authors
Comments and Suggestions for Authors
The paper is interesting. However to improve publication quality I have the following suggestions:
· Results shoud be corrected: this section should be supplemented with the results of statistical tests. Statistically significant differences in a given parameter depending on the tested sodium chloride concentration should be marked on the graphs. I suggest marking statistically insignificant values ​​with the same letters (level of significance: *p<0.05, **p<0.01, ***p<0.001).
· The Introduction section should be rewritten with particular attention paid to the aim of the work and the justification for undertaking research on Penicillium chrisogenum P13.
· Some minor linguistic correction is required.
· Lines 195-196: There are no citations here - who is the author of these results?
· Line 218: Did you mean superoxide anion radical?
Author Response
Dear Reviewer,
We appreciate all your valuable comments on our manuscript. We revised the manuscript according to the comments and suggestions. We believe that the manuscript has been further improved. You can find description of our corrections bellow.
General comments: The paper is interesting. However to improve publication quality I have the following suggestions:
Comments 1. Results shoud be corrected: this section should be supplemented with the results of statistical tests. Statistically significant differences in a given parameter depending on the tested sodium chloride concentration should be marked on the graphs. I suggest marking statistically insignificant values ​​with the same letters (level of significance: *p<0.05, **p<0.01, ***p<0.001).
Response 1. Thank you for your advice! We added statistical analysis of the results
Comment 2. The Introduction section should be rewritten with particular attention paid to the aim of the work and the justification for undertaking research on Penicillium chrisogenum P13.
Response 2. We rewrote the Introduction section and hope it is clearer and more focused now!
Some minor linguistic correction is required.
Comments 3. · Lines 195-196: There are no citations here - who is the author of these results?
Response 3. We corrected the sentence.
Comments 4. Line 218: Did you mean superoxide anion radical?
Response 4. Yes, we mean superoxide anion radical. We corrected the sentence.
Round 2
Reviewer 1 Report
Comments and Suggestions for Authors
I am satisfied with the improvement of the manuscript.
Author Response
Dear Reviewer,
thank you very much for your help and cooperation!
Best regards!
E. Krumova